# Hermetic Seal of Organic Light Emitting Diode with Glass Frit

**DOI:** 10.3390/molecules27010076

**Published:** 2021-12-23

**Authors:** Chien-Liang Chiu, Meng-Syun Lin, Yi-Chen Wu

**Affiliations:** 1Department of Electronics Engineering, National Kaohsiung University of Science and Technology, Kaohsiung 811, Taiwan; clchiu@nkust.edu.tw (C.-L.C.); baty11270116@gmail.com (M.-S.L.); 2Department of Telecommunication Engineering, National Kaohsiung University of Science and Technology, Kaohsiung 811, Taiwan

**Keywords:** organic light-emitting device (OLED), laser encapsulation, glass frit

## Abstract

The components of OLED encapsulation with hermetic sealing and a 1026-day lifetime were measured by PXI-1033. The optimal characteristics were obtained when the thickness of the TPBi layer was 20 nm. This OLED obtained a maximum luminance (Lmax) of 25,849 cd/m^2^ at a current density of 1242 mA/cm^2^, an external quantum efficiency (EQE) of 2.28%, a current efficiency (CE) of 7.20 cd/A, and a power efficiency (PE) of 5.28 lm/W. The efficiency was enhanced by Lmax 17.2%/EQE 0.89%/CE 42.1%/PE 41.9%. The CIE coordinates of 0.32, 0.54 were all green OLED elements with wavelengths of 532 nm. The shear strain and leakage test gave results of 16 kgf and 8.92 × 10^−9^ mbar/s, respectively. The reliability test showed that the standard of MIL-STD-883 was obtained.

## 1. Introduction

Optoelectronic devices based on organic materials have several advantages. They are low-cost, their power efficiency is high, and they create mechanically flexible devices. The concept of cheap solar cells providing clean green energy on a large scale is interesting. Furthermore, organic light-emitting diodes (OLEDs) are a promising technology for use in energy-efficient flexible light sources and displays [1]. A major drawback of organic electronics is their relatively poor environmental stability. Moisture and oxygen can penetrate into the organic stack through pinholes in the metal cathode layer. These pinholes are induced by particles that are present during the processing of the devices [1,2,3]. Lateral diffusion of water and oxygen in the organic stack enables cathode oxidation over a continuously growing area [4]. The improvement of the encapsulation technique, electrode materials, and substrate processing in recent years has helped to overcome the problem of OLED degradation [2]. However, organic materials are usually very susceptible to humidity, water, and oxygen, which cause damage to the organic material layer and electrode (cathode) of the device, affecting its lifetime. As a result, favorable encapsulation is indispensable [3]. A material with low moisture diffusion and absorption properties needs to be identified in order to avoid damage to the OLED [4].

The main reason why OLEDs have not completely replaced other display products is that their stability and yield are worse than in other displays. Their luminous efficiency and other characteristics have dropped sharply [4,5], so that research on the hermetic packaging process of OLED is a key issue. Many technologies have been proposed in recent years. Among them is a method developed in Japan whereby a closed cover outside the component is used to protect and isolate the outside air and moisture in the sealed cover. However, this method will increase the thickness of the device due to the shape of the sealed cover and is susceptible to physical impact and damage. In the case of large-scale manufacturing, there will be a problem due to its poor heat dissipation properties [6]. In South Korea, another technology in which a moisture absorbent is added to epoxy resin was developed. The volume expansion caused by the reaction of moisture absorbents with moisture may cause physical damage to organic components, and when metal oxides are used as moisture absorbents, they will react with moisture to produce strong alkaline substances that cause chemical damage to the organic layer or cathode [7].

In the OLED packaging process, organic materials are extremely sensitive to temperature, so that the temperature and packaging methods are extremely important [8]. Having an excessive temperature in the packaging process will lead to degradation or damage to the light-emitting characteristics and the lifetime of OLED components [9]. However, the hermetic seal of the material with traditional UV glue in the packaging process does not reach 10^−8^ Torr [10]. Thus, the packaging material currently used in the industry is glass frit. Glass frit glue is mainly made of inorganic materials. When it is used as an OLED packaging material, it can block the penetration of moisture and oxygen from the external environment [11]. It is not necessary to attach a moisture-absorbent material in the OLED package, and the device is able to exist in conditions of high temperature and high humidity (85 °C/85% RH) for more than 7000 h [12]. The glass frit glue coating can be made by using a screen-printing or dispenser method in which the glass packaging cover is coated, and then the packaging glass cover and the vapor-deposited OLED-related materials are covered in a nitrogen-filled environment. The glass glue is melted and adhered by a laser welding system.

Glass frit glue must be able to quickly absorb the energy of the laser beam and reach the melting point in a short time. In addition, the coefficient of thermal expansion (CTE) must be equivalent to that of the ITO glass substrate to avoid a deviation in the alignment package with a large difference in the CTE [13]. The laser welding method uses local heat to melt the glass frit glue; it will therefore not cause damage to temperature-sensitive organic materials.

Laser welding technology is used in encapsulation due to its coherence, non-contact processing, and complex shape processing [14,15,16]. It is not only beneficial for OLEDs with temperature-sensitive materials but also fills the requirement of blocking the water and oxygen. The usage of glass frit glue has advantages such as a relatively lower joining temperature of almost 350 °C and less rigorous requirements for contact surface smoothness [17,18,19,20].

## 2. Experiment and Devices

The light-emitting principle of OLED diodes is carrier injection. Under the basic single layer structure of an OLED device, the emitted light must pass through the device. The OLED device must have one transparent side electrode so that the emitted light can pass through the element. When a forward bias is applied to the positive and negative electrodes, the carriers will generate holes and electrons from the anode and cathode ends, respectively. The holes and electrons will be injected into the energy level of the highest occupied molecular orbitals, HOME, of the luminescent material and the energy level of the lowest unoccupied molecular orbital, LUMO. Finally, photons are generated through the principle of radiative recombination. The potential difference between the two electrodes makes the two carriers move in the organic material layer and finally recombine in the light-emitting layer. The energy released after carrier radiative recombination leads to the formation of a radiative exciton. The state of the carrier returns from the high energy level of the excited state to the low energy level of the ground state. The difference in energy is released into heat or photons, and the wavelength emitted by the device depends on the inherent fluorescent properties of the organic luminescent material. When an external bias voltage is applied, the cathode and anode individually inject electron and hole carriers toward the organic layer. The injected electrons/holes go from the electron/hole transport layer to the organic structure. As the two carriers move to the light-emitting layer, electrons and holes combine in the light-emitting layer and generate excitons. The generated excitons migrate under an applied voltage and transfer the generated energy to the light-emitting layer. This is where the excited electrons transition from the ground state to the excited state, and finally, the energy generated in the excited state is passivated back to the ground state by radiative photons generated to release light energy. The light-emitting layer has two kinds of transition. The first is fluorescence, and the second is phosphorescence. The difference between the two kinds of light transition is that the initial state of the transition of fluorescence is the singlet excited state, S1, in which the direction of the spin of the excited electron is opposite to the direction of the unexcited electron in the ground state. Phosphorescence is a triplet excited state, T1, in which the excited electron spin direction is the same as the unexcited electron direction in the ground state. Under general conditions, the S1 energy is greater than that of T1. As the transition mode that allows the carrier to directly excite it to T1 is not quite achieved, the singlet excited state (S1) has electron spin characteristics. However, it has the opportunity to transform into the triplet excited state (T1) through different transition states. Under ideal conditions, the numbers of injected holes and electrons are equal, with values of 1. Theoretically, the ratio of electrons in the singlet excited state to those in the triplet excited state is 1:3. Thus, it is generally believed that the internal quantum efficiency limitation of fluorescent materials is 25%, and the remaining 75% of the energy is non-radiated by the triplet excited state. The light extraction rate is about 1/2n^2^, where n is the refractive index. As the refractive index of the glass substrate is 1.5 and its light extraction rate is about 20%, the theoretical upper limitation of the external quantum efficiency (EQE) is about 5%.

In the experiment, we used a 0.3 × 0.3 cm^2^ active area basic un-doped OLED with a three-layered structure. The cathode consisted of aluminum (Al) deposited on lithium fluoride (LiF). The HTL consisted of *N*,*N*′-di[(1-naphthyl)-*N*,*N*′-diphenyl]-(1,1′-biphenyl)-4,4′-diamine (NPB). Alq3 acted as an emitting layer (EML). The ETL consisted of 2,2′,2″-(1,3,5-Benzinetriyl)-tris(1-phenyl-1-H-benzimidazole) (TPBi). The device structure is shown in Figure 1. All organic layers were deposited under high-vacuum conditions of 1.2 × 10^−6^ Torr, and the OLED devices were transferred directly into an automated laser processing system with 10 L/min of nitrogen (N_2_) gas for encapsulation.

The encapsulation devices had a laser power of 2.595 (W) and scanning speed of 0.1 mm/s to cure the glass frit with a melting point of 320–350 °C. To blow N_2_ gas at a rate of 10 L/min during the encapsulation process and prevent water humidity and oxygen from entering the air environment, the encapsulation area protected the organic materials.

The OLED device encapsulation procedure used was as follows. (1) The groove glass packaging cover was shaken for 10 min in an ultrasonic oscillator with DI water/pure alcohol/alcohol in order to clean it. After the cleaning process had been completed, the groove glass packaging cover was sprayed dry with N_2_ gas. The groove glass put into the petri dish was placed in the oven and heated at 60 °C for 10 min. (2) For the parameters of the dispenser, we used a dispensing speed of 1 mm/s, a dispensing time of 2 s, and a stop time of 1 s. The dispensing pressure was adjusted depending on the packaging glue. Finally, the syringe (G 30) was filled with glue with a UV glue pressure of 1 kg/cm^2^ and a glass frit glue pressure of 2 kg/cm^2^. The dispensing path was set after the parameters were set. (3) To calibrate the focal length of the laser in order to ensure that the laser was focused on the packaging glue during the encapsulation process when the automated laser packaging platform was used, a Vernier caliper was used to measure the distance between the laser and the platform (15 cm), and the focal length was adjusted with white paper to observe the minimum light point of the laser spot. The laser power was set to 0.35 W with a current of 10 A and an automated laser path. (4) After the groove glass packaging cover was dispensed with glass frit glue, the groove glass packaging cover was aligned to the OLED substrate and the laser scanning path was set. (5) The output mode and output time of the laser were chosen. The laser output mode was set to the CW mode, and the output time was matched to the laser scanning time. During the laser welding process, N_2_ gas was applied at a flow rate of 10 L/min to prevent water and oxygen from entering the component and reducing its lifetime under atmospheric conditions. Finally, the OLED component was taken out from the plastic vacuum drying vessel and the encapsulation cover was aligned to perform the laser welding encapsulation process.

## 3. Results and Discussion

### 3.1. Space Charge Limited Current, SCLC Model for the OLED Device

Under a low electric field of ≤3 × 10^5^ V/cm, the energy barrier between the organic layer and the metal interface was low and the carrier injection was reduced. The current was dominated by the conductive carriers of the organic layer. An ohmic contact was formed [21].
(1)$$J_{SCLC}=\frac{9}{8}\varepsilon_{r}\varepsilon_{0}\mu \frac{V^{2}}{d^{3}}$$

Under a greater electric field of >3 × 10^5^ V/cm, the current density measured in the experiment was greater than that determined by theoretical calculation; the carrier mobility was therefore taken into account with the change in the electric field [22].
(2)$$J_{SCLC}=\frac{9}{8}\varepsilon_{r}\varepsilon_{0}\mu \frac{V^{2}}{d^{3}}\exp\left(\beta \sqrt{E}\right)$$
where *J_SCLC_* is the current density, *ε*_0_ is the vacuum permittivity, *ε_r_* is the dielectric constant of the material, *µ* is the carrier mobility, *V* is the applied voltage, *d* is the thickness, *β* is the factor of the Poole–Frenkel effect, and *E* is the electric field.

The parameters were substituted into the SCLC theoretical model using Mathcad software. The curve of each film thickness with the current density and voltage is shown in Figure 2. The charge density of the electrons with a TPBi thickness of 10/15/20 nm gradually approached the charge density of the holes. As the charge density of the electrons was not close to the charge density of the holes and gradually increased from 10 to 20 nm, the combination rate of electrons and holes was enhanced at the thickness of 15 nm, and then there was a gradual decrease in the combination rate of electrons and holes with a thickness of 20 nm. Finally, a charge density of the electrons of 15 to 20 nm was also much greater than the charge density of the holes.

### 3.2. OLED Device Measurement

Under different TPBi film thickness of 10/15/20/30/40 nm, the electroluminescence spectra were all emitted at a wavelength of 532 nm, and the CIE coordinates were set at (0.32, 0.54), as shown in Figure 3. The OLED device was measured with the optimal characteristics when the thickness of the TPBi was 20 nm. The device had a current density of 2190 mA/m^2^ at 10 V with encapsulation and 2203 mA/m^2^ at 10 V with no encapsulation. The maximal luminance (Lmax) was 21,480 cd/m^2^ with encapsulation and 21,523 cd/m^2^ with no encapsulation. An Al cathode thickness of 100 or 150 nm did not have a significant influence, as shown in Figure 4a–d. The external quantum (EQE) was 1.39% when packing was used and 1.40% without packaging, and the current efficiency (CE) was 4.17 cd/A with packaging and 4.19 cd/A without packaging. The power efficiency (PE) was 3.07 lm/W with packaging and 3.08 lm/W without packaging, as shown in Figure 4e–h. The luminance loss with and without encapsulation was 0.2%. Figure 4k compares the numerical calculation of the current density with experimental data under different TPBi thicknesses of 10/15/20/30/40 nm. When the TPBi thickness was 10/15/20/30/40 nm, the current density was 1189/1384/2176/1226/1053 mA/m^2^ in the numerical calculation and 1114/1394/2190/1202/1067 mA/m^2^ in the experimental data. The results of the numerical calculation agree with the measured data.

The shear strain test determines the shear strength with encapsulation, i.e., the maximum shear stress that the material can withstand before failure occurs. The shear strain test with encapsulation shown in Figure 4l gave tensile strength values of 2.566 kgf for thermosetting glue, 7.348 kgf for UV glue, and 16 kgf for glass frit glue. These are all above the standard value of 10.2 kgf (MIL-STD-883).

When the thickness of TPBi was changed and calculated according to the SCLC theory, the current density at a thickness of 10 nm was obtained, and the maximum value was 10,510 mA/cm^2^. The current density at the maximum thickness of 40 nm was the minimum value of 1350 mA/cm^2^. However, at a thickness of 15 nm, there was a current density of 8178 mA/cm^2^. When the thickness was 20 nm, the current density was 9644 mA/cm^2^, and the current density was 2479 mA/cm^2^ at a thickness of 30 nm. As the SCLC theory does not take the capacitance and resistance effects of organic light-emitting diodes into account, this theory does not allow us to know the maximum breakdown voltage of the device. The effective circuit of the device and the cause of device breakdown are discussed. It could be known that the charge density of holes decreases slowly after 4 V. With an external bias voltage of 3 to 10 V, the charge density of the electrons gradually approached the charge density of the holes. As the charge density of the electrons was not close to the charge density of the holes and was gradually increasing, there was a gradual decrease in the combination rate of electrons and holes at an interface thickness of 20 nm. Eventually, the charge density of the electrons was much greater than the charge density of the holes.

### 3.3. Oxygen Plasma Bombards to ITO Thin Film Substrate

To enhance the injection of hole carriers, the ITO transparent electrode was subjected to oxygen plasma bombardment. Oxygen plasma bombarded the ITO thin film substrate for different times of 2/3/4/5/8/10 min, emitting a green light at the wavelength of 532 nm, using CIE coordinates of 0.32, 0.54. The oxygen plasma bombardment time did not affect the light wavelength emitted by the OLED device. The luminance of the OLED device with an oxygen plasma bombardment time of 3 min was optimal for ohmic contact, as shown in Figure 5. The Lmax value was 25,849 cd/m^2^ at 10 V and a current density of 1242 mA/cm^2^ with encapsulation and 25,901 cd/m^2^ at 10 V without encapsulation, and the luminance loss was only 0.2%, as shown in Figure 5a,b. The EQE was 2.28% with encapsulation and 2.29% without encapsulation, and the CE was 7.20 cd/A with encapsulation and 7.20 cd/A without encapsulation. The PE was 5.28 lm/W with encapsulation and 5.3 lm/W without encapsulation, as shown in Figure 5c–h.

### 3.4. SED Model for OLED Degradation

After hermetic sealing package of the OLED device with a lifetime of 1500 h, OLED encapsulation was carried out and measured with the NI PXI-1033. The luminance of the OLED device was 1000 cd/m^2^, and a constant current source was continuously applied. This was set as the control standard CIE 150-2003. The measured data were substituted into the stretched exponential decay (SED) model to calculate the lifetime of the component. Under the SED model of OLED degradation, the OLED luminance with respect to time is expressed as Equation (3).
(3)$$\frac{L}{L_{0}}=\exp[-{\left(\frac{t}{\tau }\right)}^{\beta }]$$
where *L* is the OLED luminance, *L*_0_ is the initial OLED luminance, *t* is the current time, *τ* is the characteristic time of decay, and *β* is a stretching exponent [7,8].

As shown in Figure 6, an OLED device burned-in with thermosetting glue obtained a decay of 0.99813 at 8 h, 0.97479 at 16 h, and 0.95358 at 24 h. An OLED device burned-in with UV glue obtained a decay of 0.99893 at 8 h, 0.98482 at 16 h, and 0.97130 at 24 h. An OLED device burned-in with glass frit glue obtained a decay of 0.99896 at 8 h, 0.99833 at 16 h, and 0.99780 at 24 h. Finally, the lifetime of the OLED device was calculated to be 20 days with thermosetting glue, 60 days with UV glue, and 1026 days with glass frit glue.

### 3.5. The Hermetic Measurement of the OLED Device

The reliability of the device’s packaging with different materials was measured by a helium leak detector, which is a small gas mass spectrometer. The standard measurement is done according to reference MIL-STD-883. When the packaging material used was glass frit glue, the leakage was 8.92 × 10^−9^ mbar/s. The leakage test aligned with MIL-STD-883(<10^−6^ mbar/sec) and was within the standard range. As the material was used UV glue, the leakage was 2.1 × 10^−5^ mbar/s or 1.4 × 10^−6^ mbar/s.

## 4. Conclusions

When the electron transport layer (TPBi) thickness was changed, the TPBi thickness of 20 nm in this OLED device obtained the optimal characteristics with the hermetic sealing package. The characteristics were a maximum luminance of 21,480 cd/m^2^ at current density of 2190 mA/cm^2^, EQE of 1.39%, CE of 4.17 cd/A and PE of 3.07 lm/W, respectively. When the OLED device first were pre-treated so that the ITO glass substrate was done by the oxygen plasma bombardment under different time, the OLED device with the oxygen plasma bombardment of 3 min had the optimal characteristics. The device had a maximum luminance of 25,849 cd/m^2^ at current density of 1242 mA/cm^2^ with the luminance enhanced by 17.2%, EQE of 2.28% with EQE enhanced by 0.89%, CE of 7.20 cd/A with CE enhanced by 42.1% and PE of 5.28 lm/W with PE enhanced by 41.9%, respectively. The International Commission on Illumination (CIE) of chromaticity coordinates of OLED devices were all 0.32, 0.54. The OLED device is a green light at the wavelength of 532 nm.

Under the package test, the three different materials of thermosetting glue, UV glue and glass frit glue were used. The glass frit glue material of the encapsulation was cured for the continuous wave laser (CW laser) at the wavelength of 800 nm. To apply local heating of the favorable characteristic of the CW laser with the encapsulation could reduce the influence of temperature for organic materials. The CW laser power of 2.595 W and scanning speed of 0.1 mm/s were used to cure the glass frit glue for OLED hermetic sealing encapsulation. The shear strain test was 16 kgf and the leakage test was 8.92 × 10^−9^ mbar/s. The glass frit glue under the reliability test had reached the standard of MIL-STD-883. i.e., A hermetic seal package standard was only with glass frit glue. The hermetic seal of the OLED device was achieved for 1026 days of lifetime measured by PXI-1033 under the TPBi thickness of 20 nm and oxygen plasma bombardment of the ITO glass substrate for 3 min.

## Figures and Tables

**Figure 1 molecules-27-00076-f001:**
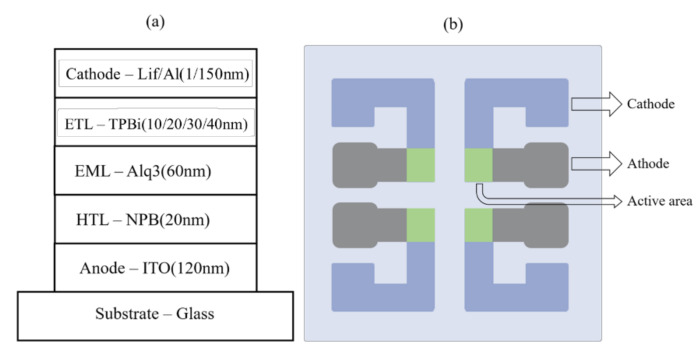
The device structure employed as a (**a**) cross section, (**b**) schematic diagram.

**Figure 2 molecules-27-00076-f002:**
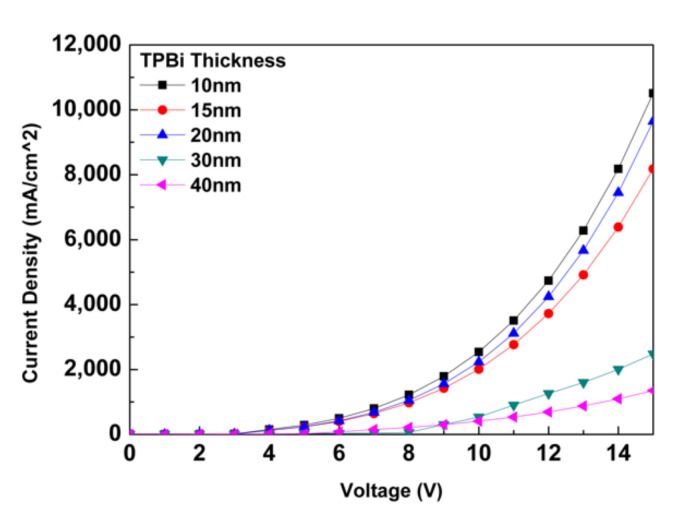
The current density and voltage curve diagram determined by the numerical calculation of the SCLC theoretical model.

**Figure 3 molecules-27-00076-f003:**
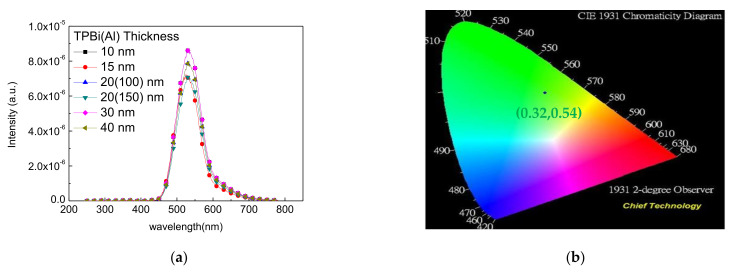
(**a**) The electroluminescence spectra under different TPBi film thicknesses, (**b**) CIE coordinates.

**Figure 4 molecules-27-00076-f004:**
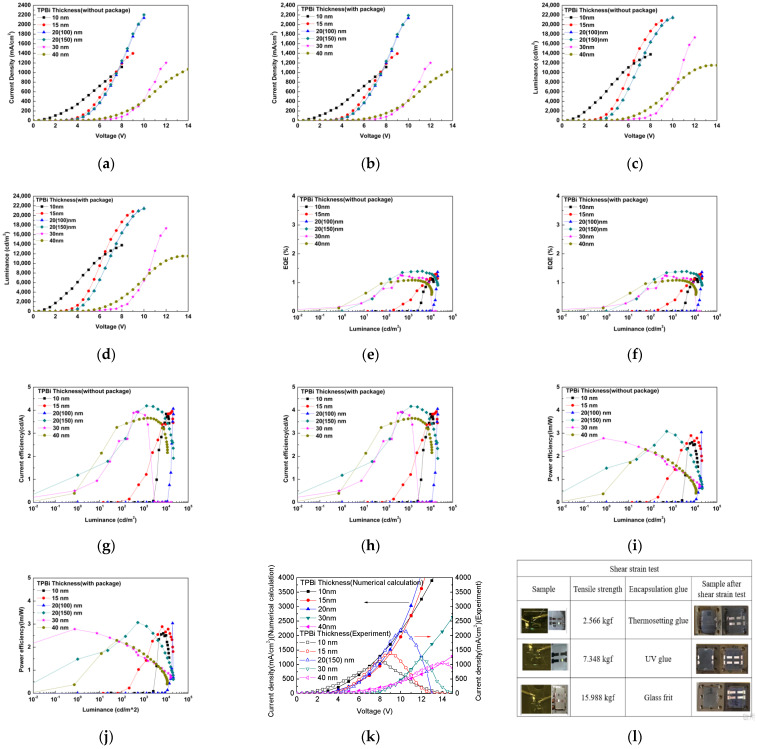
Results under the different TPBi thicknesses for the current density and voltage (**a**) without packaging and (**b**) with packaging; luminance and voltage (**c**) without packaging and (**d**) with packaging; EQE and luminance (**e**) without packaging (**f**) and with packaging; the current efficiency and luminance (**g**) without packaging and (**h**) with packaging; power efficiency and luminance (**i**) without package (**j**) and with packaging (**k**); and the current density and voltage in the numerical calculation and experiment. (**l**) Shear strain test for the OLED device with encapsulation.

**Figure 5 molecules-27-00076-f005:**
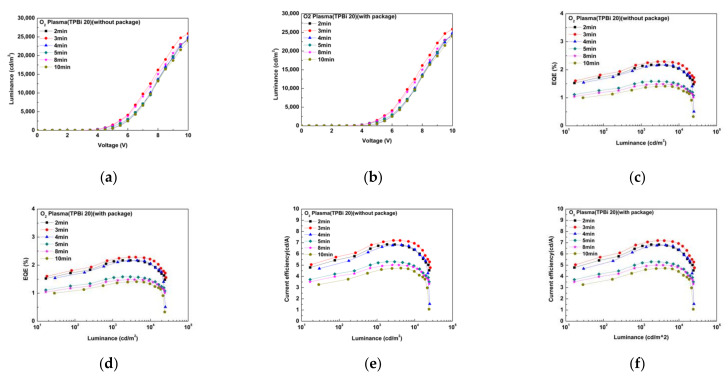
Results under the different oxygen plasma bombardment times for luminance and voltage (**a**) without packaging and (**b**) with packaging; EQE and luminance (**c**) without packaging and (**d**) with packaging; current efficiency and luminance (**e**) without packaging and (**f**) with packaging; and power efficiency and luminance (**g**) without packaging and (**h**) with packaging.

**Figure 6 molecules-27-00076-f006:**
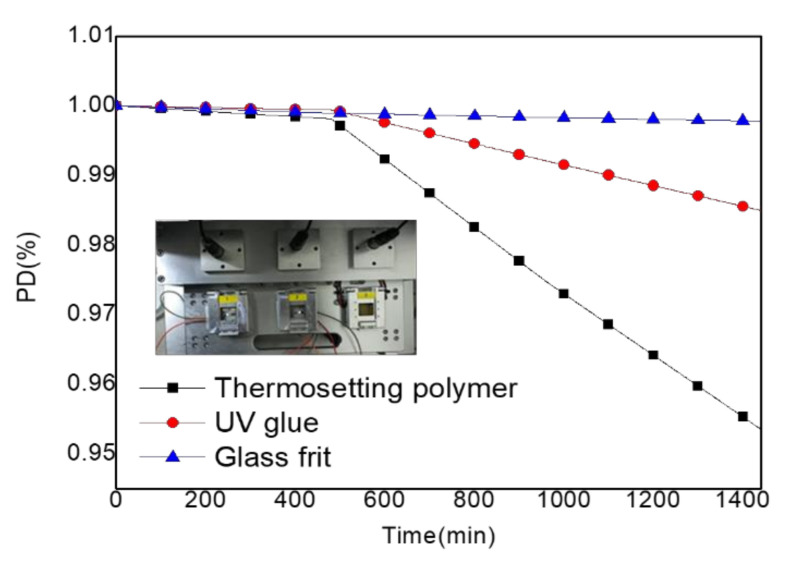
OLED device burned-in with PD% and time.

## Data Availability

Not applicable.

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
