# Peer review of "Hermetic Seal of Organic Light Emitting Diode with Glass Frit"

_molecules, 2021, doi:10.3390/molecules27010076_

Round 1

Reviewer 1 Report

The authors demonstrated encapsulation OLED by using the material of glass frit, and discussed the optical and electrical characteristics of the OLED with and without package under different TPBi film thickness as well as different bombardment time of the oxygen plasma for ITO glass substrate. Under the encapsulation testing, the reliability of OLED by using the material of glass frit showed optimum and the luminance loss was low. However, there are a few questions to be addressed before moving forward.

  1. The proper references should be added for equation 1 and 2.
  2. Can the authors add discussions about why the optimal characteristics can be obtained when the thickness of the TPBi is 20 nm. Also, the oxygen plasma bombardment time of 3 min for ITO glass substrate is the optimal condition.
  3. The resolution of fig.4 and fig.5 need to be adjusted.

Author Response

To the comments by Referee #1:

  1. The proper references should be added for equation 1 and 2.

    Equation 1 and 2 are already added for reference 21 & 22 The changes are marked in red color.

  2. Can the authors add discussions about why the optimal characteristics can be obtained when the thickness of the TPBi is 20 nm. Also, the oxygen plasma bombardment time of 3 min for ITO glass substrate is the optimal condition.

    But this SCLC theory does not take into account the capacitance and resistance effects of organic light-emitting diodes, so this theory does not know the maximum breakdown voltage of the device. The effective circuit of the device and the cause of the device breakdown will be discussed. It can be known that the charge density of holes decreases in a slow trend after 4 V. With an external bias voltage of 3 V to 10 V, the charge density of electrons gradually approaches the charge density of holes. As the charge density of electrons is not close to the charge density of holes and gradually increased, so resulting in a gradual decrease in the combination rate of electrons and holes at the 20 nm interface, and finally the charge density of electrons will be too much greater than the charge density of holes. The changes are marked in red.

    To enhance the injection of hole carriers, the ITO transparent electrode is done by oxygen plasma bombardment. the oxygen plasma bombardment time of 3 min for ITO glass substrate is obtained the optimal Ohmic contact. The changes are marked in red.

  3. The resolution of fig.4 and fig.5 need to be adjusted.

    The resolution in Fig. 4 and Fig. 5 have already modified.

    Sincerely,

    Yi-Chen Wu * & Chien-Liang Chiu
    Department of Electronics Engineering
    National Kaohsiung University of Science and Technology E-mail: 72325@nkust.edu.tw & clchiu@nkust.edu.tw

Reviewer 2 Report

Reviewer report:

C.L Chiu et. al. reported the research article that claims “Hermetic Seal of Organic Light Emitting Diode with Glass Frit and lifetime of 1026 days.” The authors should clearly mention the novelty of the results by comparing them with previously reported similar works. The significance of the work is not very high enough to publish in Molecules. Few papers with better device performance have been reported using Laser fit sealing methods. This work is only the optimization of the methodology, and the OLED performance is not exciting, which will abate the novelty of the manuscript. The authors further need to improve the manuscript before submitting it by considering the following points.

1. Abstract: page 1 line10 contains typo errors, EQE abbreviation should be “external quantum efficiency”.

2. Introduction: This should be checked carefully because few sentences are repetitive in meaning.

Page1 line 22 “Furthermore, organic light-emitting diodes (OLED) are a promising technology for energy-efficient flexible light sources and displays.” Looks similar to Page 1 line 28 “OLED devices have attracted much attention due to their potential value in a number of lighting and display in the future [1].”

Page1 line 31 “However, the organic materials are usually very susceptible to humidity water and oxygen.” Looks similar to Page 1 line 37 “The organic materials used in OLED devices are susceptible to the intrusion of oxygen and humidity water. The organic materials that are extremely sensitive to water and oxygen cause damage to the organic.”

Use any one form of the words “glass glue” or “glass frit glue”

3. There are a few unclear sentences that need to be reframed.

Page3 line 94 “To use 10 L/min of N2 gas under the encapsulation process to reduce water humidity and oxygen in an air environment.”

Page 3 line 117 “To avoid the component and package damage under atmospheric conditions.”

4. Page 4 lines 133 and 134 need to be clarified “JSCL or JSCLC

5. in Figure 2. The author needs to explain the current density and voltage deviation of 20 nm thickness compared with15 nm.

6. The author clearly explains the motivation of the experiment rather than directly jumping to the results.

Page 4 line 161, it is better to write a few sentences in the manuscript about the importance and purpose of the shear strain test.

And also on page 5 line 178, it is better to write a few sentences in the manuscript about the importance and purpose of oxygen plasma bombardment.

7. Figure 4 and figure 5 due to very poor visibility it’s hard to read - authors should plot this such that the graphs are more legible. Please provide better visualization of data (matching the order of the annotation along with the better visibility of captions, etc.)

8. Conclusion looks similar to OLED measurement discussion and the majority of the data should be moved into the experimental section part. The author needs to clearly conclude his findings in this section.

In-Page 7 line 236 “The International Commission on Illumination (CIE) of chromaticity coordinates of OLED devices are all (0.32, 0.54). The OLED device is a green light at the wavelength of 532 nm”. The above sentence has a similar meaning on Page 7 line 250 “The CIE coordinate of (0.32, 0.54) is a green OLED element with a wavelength of 532 nm.”

9. In Page 7 line 250, it needs to be clarified how the EQE is raised to 11.42 %.
10. There are some other mistakes in this manuscript. There are writing mistakes in Line 10 on Page 1, Line 165 on Page 5, Line 250 on Page 8.

Author Response

To the comments by Referee #2:

  1. Abstract: page 1 line10 contains typo errors, EQE abbreviation should be “external quantum efficiency”.
    Thank you for referee. It is already correction. The change is marked in red.

  2. Introduction: This should be checked carefully because few sentences are repetitive in meaning.
    (a)Page1 line 22 “Furthermore, organic light-emitting diodes (OLED) are a promising technology for energy-efficient flexible light sources and displays.” Looks similar to Page 1 line 28 “OLED devices have attracted much attention due to their potential value in a number of lighting and display in the future [1].”

    “LED devices have attracted much attention due to their potential value in a number of lighting and display in the future [1]” is withdrawn.
    (b)Page1 line 31 “However, the organic materials are usually very susceptible to humidity water and oxygen.” Looks similar to Page 1 line 37 “The organic materials used in OLED devices are susceptible to the intrusion of oxygen and humidity water. The organic materials that are extremely sensitive to water and oxygen cause damage to the organic.”

    “The organic materials used in OLED devices are susceptible to the intrusion of oxygen and humidity water. The organic materials that are extremely sensitive to water and oxygen cause damage to the organic.” is withdrawn.“

    However, the organic materials are usually very susceptible to humidity water and oxygen.” is added “cause damage to the organic material layer and electrode (cathode) of the device, which affects the lifetime of the OLED device.” The changes are marked in red.

         (c)Use any one form of the words “glass glue” or “glass frit glue”

          It would be unified to glass frit glue. The changes are marked in red.

  1. There are a few unclear sentences that need to be reframed.

    • (a)Page3 line 94 “To use 10 L/min of N2 gas under the encapsulation process to reduce water humidity and oxygen in an air environment.”

      I am thankful of your assistant. The sentence is modified to “To blow 10 L/min of N2 gas under the encapsulation process to prevent water humidity and oxygen in an air environment” The changes are marked in red.

    • (b)Page 3 line 117 “To avoid the component and package damage under atmospheric conditions.”

             The sentence is modified to “To avoid water and oxygen into the  

             componentand the damage of lifetime under atmospheric conditions”      

             The changes are marked in red.
  2. Page 4 lines 133 and 134 need to be clarified “JSCL or JSCLC”

         It is already correction for JSCLC.

    5.  in Figure 2. The author needs to explain the current density and

         voltage deviation of 20 nm thickness compared with15 nm.

The charge density of electrons in thickness TPBi of 10 nm/ 15 nm/ 20 nm gradually approaches the charge density of holes. As the charge density of electrons is not close to the charge density of holes and gradually increased from 10 nm to 20 nm, so the combination rate of electrons and holes is enhanced in the thickness of 15 nm and then a gradual decrease in the combination rate of electrons and holes in thickness of 20 nm. Finally, the charge density of electrons from 15 nm to 20 nm will be too much greater than the charge density of holes. The changes are marked in red.

6.The author clearly explains the motivation of the experiment rather than directly jumping to the results.

  • Page 4 line 161, it is better to write a few sentences in the manuscript about the importance and purpose of the shear strain test.

    A shear strain test is to determine the shear strength with encapsulation, which is the maximum shear stress that the material can withstand before failure occurs of a material. The changes are marked in red.

  • And also on page 5 line 178, it is better to write a few sentences in the manuscript about the importance and purpose of oxygen plasma bombardment. To enhance the injection of hole carriers, the ITO transparent electrode is done by oxygen plasma bombardment. the oxygen plasma bombardment time of 3 min for ITO glass substrate is obtained the optimal Ohmic contact. The changes are marked in red.

7. Figure 4 and figure 5 due to very poor visibility it’s hard to read - authors should plot this such that the graphs are more legible. Please provide better visualization of data (matching the order of the annotation along with the better visibility of captions, etc.).

The resolution in Fig. 4 and Fig. 5 have already modified.  

8. Conclusion looks similar to OLED measurement discussion and the majority of the data should be moved into the experimental section part. The author needs to clearly conclude his findings in this section.

Yes. Conclusion would be modified. The changes are marked in red.  

In-Page 7 line 236 “The International Commission on Illumination (CIE) of chromaticity coordinates of OLED devices are all (0.32, 0.54). The OLED device is a green light at the wavelength of 532 nm”. The above sentence has a similar meaning on Page 7 line 250 “The CIE coordinate of (0.32, 0.54) is a green OLED element with a wavelength of 532 nm.

The sentence “The CIE coordinate of (0.32, 0.54) is a green OLED element with a wavelength of 532 nm.” is withdrawn.  

9. In Page 7 line 250, it needs to be clarified how the EQE is raised to 11.42 %.

Sorry. The data of EQE/CE/PE are not correction. They are typed wrongly. These are modified to EQE of 2.28%, CE of 7.2 cd/A and PE of 5.28 lm/W. The changes are marked in red.  

10. There are some other mistakes in this manuscript. There are writing mistakes in Line 10 on Page 1, Line 165 on Page 5, Line 250 on Page 8.

These are correction. The changes are marked in red.

Sincerely,

 Yi-Chen Wu * & Chien-Liang Chiu

 Department of Electronics Engineering

 National Kaohsiung University of Science and Technology

 E-mail: 72325@nkust.edu.tw & clchiu@nkust.edu.tw  

Reviewer 3 Report

The manuscript presents investigations in field of OLED devices. English of the paper should be very strongly improved by specialist of the field before review of the paper.

-Abstract of the paper in not homogeneous and is not suitable for a scientific journal.

-English of the paper is so strange and complicated that it is very difficult to understand the paper ?

I recommend correction of English firstly and then re-submission of the manuscript.

Author Response

To the comments by Referee #3:

  1. Abstract of the paper in not homogeneous and is not suitable for a scientific journal. The referee does not clear indicate about abstract issue. The authors will be modified it as soon as possible. The changes are marked in red.
  2. English of the paper is so strange and complicated that it is very difficult to understand the paper? The article is revised in English. The changes are marked in red.

Sincerely,

                                   Yi-Chen Wu * & Chien-Liang Chiu

                                   Department of Electronics Engineering

                                   National Kaohsiung University of Science and Technology

                                   E-mail: 72325@nkust.edu.tw & clchiu@nkust.edu.tw

Round 2

Reviewer 2 Report

Revision review report: I think the revised version of the manuscript entitled “Hermetic Seal of Organic Light Emitting Diode with Glass Frit” can be published after correcting the minor typo errors as given bellow

  1. Throughout the manuscript, the author needs to check the similarity in the representation of units

For example, “85%” and “85 %”, 2.28% and 2.29 % without and with space need to be unified.

Also, 000C, 00°C, 00 0C, 00 °C, 000 C, 00° C should be unified accordingly.

One form of “min” or “minutes” can be used.

  1. In the experimental section, 3rd paragraph the word “respective” need to be corrected to “respectively”.

         “Laser” in the middle of the sentence could be corrected to “laser”

  1. In conclusion, repeated words in the sentence to be corrected

      “The shear strain test is 16kgf and the and the leakage”

The contents of the following sentence were repeated trice in the conclusion itself. Please remove the lines

“So the optimal characteristics of this OLED device is obtained with the TPBi thickness of 20 nm and the oxygen plasma bombardment of ITO glass substrate for 3 min.”

The contents of the following sentence were repeated twice in the conclusion itself. Please remove the following sentence.
“This OLED is obtained a maximum luminance of 25849 cd/m2 at current density 1242 mA/cm2, EQE of 2.28%, CE of 7.2 cd/A and PE of 5.28 lm/W, respectively. “

Author Response

Following are the authors’ response to the referee’s comments:

To the comments by Reviewer 2:

  1. Throughout the manuscript, the author needs to check the similarity in the representation of units. “For example, “85%” and “85 %”, 2.28% and 2.29 % without and with space need to be unified. Also, 000C, 00°C, 00 0C, 00 °C, 00C, 00° C should be unified accordingly. One form of “min” or “minutes” can be used.”

They are already correction. The changes are marked in red.

  1. In the experimental section, 3rd paragraph the word “respective” need to be corrected to “respectively”. “Laser” in the middle of the sentence could be corrected to “laser”

They are correction. The changes are marked in red.

  1. (a)In conclusion, repeated words in the sentence to be corrected “The shear strain test is 16kgf and the and the leakage”

Thank you. To remove “and the”

(b)The contents of the following sentence were repeated trice in the conclusion itself. Please remove the lines. So the optimal characteristics of this OLED device is obtained with the TPBi thickness of 20 nm and the oxygen plasma bombardment of ITO glass substrate for 3 min.”

Yes. To remove this sentence.

(c)The contents of the following sentence were repeated twice in the conclusion itself. Please remove the following sentence. This OLED is obtained a maximum luminance of 25849 cd/m2 at current density 1242 mA/cm2, EQE of 2.28%, CE of 7.2 cd/A and PE of 5.28 lm/W, respectively. “

Yes. To remove this sentence.

Reviewer 3 Report

Extensive editing of English language and style required

Author Response

Following are the authors’ response to the referee’s comments

To the comments by Reviewer 3:

  1. Extensive editing of English language and style required.

Thank you for reviewer’s comments and suggestions. We would be modified them as soon as possible. The article was revised in English and style. The changes are marked in red.
